# *Satureja montana* L. and *Origanum majorana* L. Decoctions: Antimicrobial Activity, Mode of Action and Phenolic Characterization

**DOI:** 10.3390/antibiotics9060294

**Published:** 2020-05-31

**Authors:** Fernanda Gomes, Maria Inês Dias, Ângela Lima, Lillian Barros, Maria Elisa Rodrigues, Isabel C.F.R. Ferreira, Mariana Henriques

**Affiliations:** 1CEB, Centro de Engenharia Biológica, LIBRO–Laboratório de Investigação em Biofilmes Rosário Oliveira, Universidade do Minho, 4710-057 Braga, Portugal; pg33053@alunos.uminho.pt (Â.L.); elisarodrigues@deb.uminho.pt (M.E.R.); mcrh@deb.uminho.pt (M.H.); 2Centro de Investigação de Montanha (CIMO), Instituto Politécnico de Bragança, Campus de Santa Apolónia, 5300-253 Bragança, Portugal; maria.ines@ipb.pt (M.I.D.); lillian@ipb.pt (L.B.); iferreira@ipb.pt (I.C.F.R.F.)

**Keywords:** *Satureja montana*, *Origanum majorana*, decoction, antimicrobial activity, mechanism of action, phenolic characterization

## Abstract

Medicinal and aromatic plants are known to have a wide range of uses and health benefits, and should be exploited for their bioactivity. Here we evaluated the antimicrobial activity of decoctions of *Satureja montana* L. and *Origanum majorana* L. against Gram-positive and Gram-negative bacteria and *Candida* spp. as well as their mechanism of action and phenolic characterization. The *Satureja montana* and *Origanum majorana* extracts were effective against a broad set of species, including the Gram-positive *Staphylococcus aureus*, *Enterococcus faecalis,* and *Streptococcus dysgalactiae* and the Gram-negative *Klebsiella pneumoniae* and *Pseudomonas aeruginosa*. Both extracts were found to have rosmarinic acid as the main phenolic compound and to exert their antimicrobial activity at the level of the cell membrane. Membrane perturbations by the extracts impaired cell membrane integrity only a few hours after exposure. This study confirms the bioactive potential of *Satureja montana* and *Origanum majorana* decoctions, and supports the development of novel formulations with wide antimicrobial properties based on these medicinal and aromatic herbs.

## 1. Introduction

Vegetables, grains, and fruits are a good source of bioactive molecules, such as flavonoids and phenolic acids [1,2]. In plants, phenolic compounds are part of the natural defense system against several pathogens, such as bacteria, fungi, viruses, and other pests, and also play an important role in the regulation of plant hormones [3]. Polyphenols (phenolic compounds with more than one aromatic ring) can be found in the vast majority of aromatic plants, and have been associated with the plant’s bioactive properties [4]. Medicinal plants and their phenolic compounds are thus strong candidates for the development of new drugs [5], having the advantages of worldwide availability, low price, and safety [6,7]. In fact, auto-medication with plants to cure diseases and alleviate pain was a recurrent practice from the beginning of human civilization [8,9]. Now, the rising levels of resistance to traditional antimicrobial therapies are widely recognized as a global issue and require the urgent development of new approaches to fight microbial infections. In this context, extracts from plant origin are a promising antimicrobial alternative worthy of further exploration [8,9,10].

*Satureja montana* L. and *Origanum majorana* L. are annual, medicinal, and aromatic plant species from the Lamiaceae family. *Origanum majorana* is a perennial plant originally native of the Mediterranean region but widely cultivated in many countries. *Satureja montana* is a plant native from the Mediterranean region and cultivated all over Europe, Russia, and Turkey [11,12]. Both are widely used in the Mediterranean diet, holding known anti-inflammatory, antibacterial, antifungal, and antioxidant properties [12,13,14,15,16]. Here we evaluated the antimicrobial activity of *Satureja montana* L. and *Origanum majorana* L. by studying their decoction extract against several Gram-positive and Gram-negative bacteria and *Candida* spp. The mechanisms of action and the phenolic compound composition of both plant decoctions were also characterized.

## 2. Results

### 2.1. Antimicrobial Activity of Satureja montana L. and Origanum majorana L. Decoctions

Decoctions of *Satureja montana* L. and *Origanum majorana* L. were screened for their antimicrobial activities against several Gram-positive and Gram-negative bacterial spp. and *Candida* spp. using the disc diffusion test (Table 1). Strong antimicrobial effects were observed for both plants against *Staphylococcus aureus*, *Enterococcus faecalis*, and *Klebsiella pneumoniae* and for *Origanum majorana* against *P. aeruginosa*. Moderate inhibitory activity was evidenced for both plant extracts against *Streptococcus dysgalactiae* and *Satureja montana* against *Pseudomonas aeruginosa* (Table 1). In general, these extracts were effective against both Gram-negative and Gram-positive bacteria, with the exception of *Escherichia coli.* The activity of the plant decoctions against *Candida* spp. was more limited, with only *Satureja montana* showing moderate activity against *Candida tropicalis* (Table 1).

The minimum inhibitory concentration (MIC) and minimum bactericidal/fungicidal concentrations (MBC/MFC) of the plant decoctions were determined for the bacterial (one Gram-positive and one Gram-negative bacteria) and *Candida tropicalis* against which the highest antimicrobial activity was observed (Table 1). *Satureja montana* and *Origanum majorana* decoctions presented MIC and MBC values of 1.56 mg/mL (Table 2) against *S. aureus* and *K. pneumoniae*, suggesting that these extracts possess bactericidal features. *Satureja montana* decoction presented MIC and MFC values of 6.25 mg/mL, exhibiting a lower antimicrobial effect comparatively to both bacteria tested.

### 2.2. Mechanism of Action of Satureja montana and Origanum majorana Decoctions

Flow cytometry and fluorescence microscopy techniques are usually used to determine the responses of cells to antimicrobial agents and to elucidate their mechanism of action. Live/dead staining, a dual staining procedure using SYTO9 and propidium iodide, allow the differentiation of live (green) and dead (red) cells, based on cell membrane integrity. Thus, the mechanism of action, and more specifically, the effect on cell viability and membrane integrity, of *Satureja montana* and *Origanum majorana* decoctions on *S. aureus* cells was evaluated, using a live/dead assay, by flow cytometry (Figure 1) and fluorescence microscopy (Figure 2). The extracts at their MIC (1.56 mg/mL) caused a considerable decrease in live cells and an increase in damaged cells (*p* < 0.05) after 3 h of exposure. After 5 and 24 h, similar results were obtained for both extracts and control, with most of the cell population being undamaged (Figure 1 and Figure 2).

### 2.3. Identification and Quantification of Phenolic Compounds

Data obtained from the High Performance Liquid Chromatography system coupled to a Diode Array and Mass Spectrometer detectors (HPLC-DAD-ESI/MS; retention time, λmax, pseudomolecular ions, main fragment ions in MS^2^, tentative identification, and quantification) of *Satureja montana* and *Origanum majorana* decoction extracts are shown in Table 3. Twenty-four phenolic compounds were identified, of which fifteen are flavonoids (luteolin, apigenin, and quercetin derivatives), and nine are phenolic acids derivatives (*p*-coumaroyl, *p*-hydroxybenzoic, and caffeic acid derivatives: rosmarinic, salvianolic, and lithospermic acid). Interestingly, only six of the compounds identified are common to both plant extracts. While flavonoid compounds are present in higher numbers, phenolic compounds are present in higher quantity, mainly due to the high prevalence of rosmarinic acid (32% in *Satureja montana* and 56% in *Origanum majorana*).

## 3. Discussion

The antimicrobial potential of plant species has been explored mostly via studies of their essential oils (EOs) [12,13,22,23,24,25,26,27,28,29,30,31,32,33,34]. Due to the toxic effect exhibited by EOs, it is of utmost importance to explore other plant-derived therapeutic alternatives. Several works report that solubility and bioactivity of plant-derived chemical compounds greatly depend on the extraction method used to obtain the compound. Decoction has proven to be one of the best methods for the efficient extraction of bioactive compounds, and the resultant plant extracts have been suggested as a potential substitute for EOs [9,19]. Here we evaluated the bioactivity of decoctions of *Satureja montana* and *Origanum majorana* compounds against several human pathogens. To the best of our knowledge, the bioactivity of these plant decoctions has not yet been evaluated. Aqueous extracts obtained by decoctions of *Satureja montana* and *Origanum majorana* demonstrated clear antimicrobial activity against Gram-negative and Gram-positive bacteria, with the exception of *E. coli*, and limited activity against *Candida* spp., with only *C. tropicalis* responding to treatment. These results are in accordance with previous studies with extracts of these plants obtained with other methods. Methanolic extracts of both *Satureja montana* [24] and *Origanum majorana* [27] presented a broad spectrum antimicrobial activity against microorganisms, such as *S. aureus*, *E. coli*, and *C. albicans*. Ćetković et al. tested several extracts of *Satureja montana* (petroleum ether, chloroform, ethyl acetate, and *n*-butanol) against pathogens, such as *P. aeruginosa*, *E. coli*, and *S. aureus* observing that *Satureja montana* presented antibacterial activity depending on the solvent used and that *E. coli* was not sensitive to any of the extracts studied, similar to what we observed here [12]. Mohamed et al. [31] saw strong in vitro activities of ethanol and water extracts of *Origanum majorana* against *B. cereus*, *E. coli*, *S. enteritidis,* and *S. aureus*. Crude extracts of *Origanum majorana* obtained using different polarity solvents (petroleum ether, dichloromethane, ethyl acetate, and aqueous fractions) also had good results against *E. coli* and *K. pneumoniae* [29].

MIC and MBC results from this work were very similar to those previously reported by Busatta et al. [28] and Miladi et al. [34] with *Origanum majorana* (MIC = 0.92 and 0.782 mg/mL against *K. pneumoniae* and *S. aureus,* respectively) and *Satureja montana* EOs (MIC/MBC = 0.78 mg/mL against *S. aureus*). The MIC and MBC of *Satureja montana* obtained for *Candida tropicalis* were higher (6.25 mg/mL), demonstrating a weaker antifungal effect of this extract. Thus, *Satureja montana* and *Origanum majorana* decoction extracts were very effective, exerting an inhibitory effect against almost all bacteria tested and can be considered promising antibacterial agents, since decoction extracts, at 1.56 mg/mL, completely inhibited the growth of *S. aureus* and *K. pneumoniae*.

The antimicrobial effect of *Satrueja montana* and *Origanum majorana* decoctions was evaluated over time by flow cytometry aiming to determine their possible mechanism of action on *S. aureus* ATCC 25923. *S. aureus* cells showed increased permeability to propidium iodide (PI) after 3 h exposure to both decoctions, demonstrating loss of membrane integrity. Membrane damage can lead to compromised cellular homeostasis and, consequently, to the disturbance of normal cell function [35]. However, the effect on *S. aureus* membrane seemed to be reversible, since no effect on membrane integrity was observed after longer exposures. This can result from the cell’s capacity to repair some membrane lesions after injury and adapt to re-acquire their functionality by restoring their internal homeostasis and preventing cell death [36]. To increase and prolong the inhibitory effect of the decoctions, a new dose of plant extracts may be necessary after the first 3 h of treatment. By fluorescent microscopy, we observed some cell aggregation after 3 and 5 h of exposure to the plant extracts. Such effect was also reported by Annuk et al. [37] when using aqueous extracts of bearberry and cowberry leaves against *Helicobacter pylori*, which was justified by a modulatory effect on cell surface hydrophobicity. Another study observed the same aggregation effect on *E. coli* but found it to be unrelated to the antimicrobial activity of plant extracts [38]. We also observed that cell aggregation of *S. aureus* did not interfere positively with the bioactivity of *Satureja montana* and *Origanum majorana* decoctions since cells were able to restore their membrane integrity and, consequently, their functionality after exposure to both plant extracts. Overall, flow cytometry and fluorescence microscopy have shown that both plant extracts caused fast and reversible damages to the cell membranes of *S. aureus*. Therefore, although the specific mechanism of action is not yet well understood, it is possible that decoctions of *Satureja montana* and *Origanum majorana* are able to target the cell membranes by perturbing the cell membrane structures and architecture, especially in the first moments upon exposure.

The phenolic content of *Satureja montana* and *Origanum majorana* decoction extracts was also characterized in this study. Rosmarinic acid was found to be the main phenolic compound in both decoctions (36.3 mg/g in *Satureja montana* and 52.4 mg/g in *Origanum majorana*). This compound was also previously described as the main phenolic compound in *Satureja parvifolia* from India [20], *Origanum majorana* from Ireland [15], and *Satureja parvifolia* (Phil.) Epling [20]. Rosmarinic acid and caffeic acid derivatives have been extensively described in other Lamiaceae plant species [8,9,19,21]. Other caffeic acid derivatives were found in our decoctions, such as salvianolic and lithospermic acid derivatives. These derivatives were previously reported in *Origanum majorana* by Taamalli et al. [18] and *Salvia officinalis* (from the Lamiaceae family) by Martins et al. [19]. Additionally, we found salvianolic acid A and possibly lithospermic acid A present in our samples. While described in other Lamiaceae species, such as *Melissa officinalis* L. by Barros et al. [21] and *Thymus vulgaris* L. [8], this is, to the best of our knowledge, the first report of these compounds in *Satureja montana* and *Origanum majorana*. In terms of phenolic compounds, of all the caffeic acid derivatives mentioned, only two were detected in the decoction preparation of *Origanum majorana*. Although rosmarinic acid appears as the main phenolic compound in both samples, a greater variety of flavonoid compounds was identified in both samples. Three main aglycones were found, specifically luteolin, apigenin, and quercetin, which is in agreement with previous reports in other *Satureja* species, such as *Satureja hortensis* [16], *Origanum majorana* [17,18], and other Lamiaceae species [8,9,19,21]. Several luteolin derivatives were identified, among which we highlight luteolin-*O*-di-glucuronide as it is the first time it was found in *Satureja montana* and *Origanum majorana*, although previously described in other Lamiaceae species, such as *Salvia officinalis* and *Thymus vulgare* [8,19]. Finally, quercetin-*O*-glucuronide was the only quercetin derivative present in the studied samples, and here reported for the first time in *Satureja montana*.

## 4. Materials and Methods 

### 4.1. Preparation of Plant Extracts

The dry leaves of *Satureja montana* L. (winter savory) and *Origanum majorana* L. (marjoram) were obtained from “Cantinho das Aromáticas”, an organic farmer from Vila Nova de Gaia (Portugal), and further powdered (~20 mesh) before the extraction procedure. Decoctions were performed by adding 200 mL of distilled water to the sample (1 g), heating (heating plate, VELP scientific), and boiling for 5 min. The mixtures were left to stand for 5 min and then filtered under reduced pressure. The decoctions were frozen and further lyophilized (FreeZone 4.5, Labconco, Kansas City, MO, USA) to obtain a dry extract.

### 4.2. Evaluation of Antimicrobial Activity

#### 4.2.1. Disc Diffusion Assay

The lyophilized decoctions were re-dissolved in water to obtain a stock solution of 50 mg/mL. To evaluate the antimicrobial activity, different reference strains of bacteria and fungi were used, including the Gram-positive species *S. aureus* (ATCC 25923), *E. faecalis* (CECT 184), and *Streptococcus*
*dysgalactiae* subsp. equisimilis (CECT 926); the Gram-negative species *E. coli* (ATCC 25922)*, K. pneumoniae* (ATCC 11296), and *P. aeruginosa* (ATCC 10145); and the *Candida* species *C. albicans* (SC 5413), *C. tropicalis* (ATCC 750), *C. glabrata* (ATCC 2001), and *C. parapsilosis* (ATCC 20019). The antimicrobial effect was evaluated using the disc diffusion halo test. All bacterial strains were inoculated into 15 mL of Tryptic Soy Broth (TSB) from Tryptic Soy Agar (TSA) plates not older than 2 days, and grown for 24 h at 37 °C in an orbital shaker at 120 rpm. Cells were harvested by centrifugation (for 5 min at 9500× *g* at 4 °C), re-suspended in TSB, adjusted to an optical density (640 nm) equivalent to 1 × 10^6^ cells/mL, and then used in the subsequent assays. *Candida* spp. were inoculated in Sabouraud Dextrose Broth (SDB) liquid medium and incubated at 37 °C and 120 rpm during 24 h, and then adjusted to a cellular concentration of 1 × 10^5^ cells/mL. An aliquot of each strain (300 µL) was spread in TSA and Sabouraud Dextrose Agar (SDA) plates to allow for the growth of bacteria and fungi, respectively. An aliquot (25 µL) of each plant extract at 50 mg/mL was placed on a sterile blank disc. Sterile water was used as a negative control. The disks were placed on top of the bacteria and fungi plates, and the plates were incubated at 37 °C during 24–48 h. The inhibitory effects were determined by measuring the corresponding zones of inhibition (diameter of the halo of inhibition). The results were converted in a semi-quantitative scale: (-) absence of halo, (+) weak halo (3–7 mm), (++) moderate halo (8–10 mm), and (+++) strong halo (greater than 11 mm). Experiments were repeated three times.

#### 4.2.2. Determination of MIC and MBC/MFC 

MIC values of *Satureja montana* and *Origanum majorana* decoction extracts were determined by microbroth dilution technique, against *S. aureus*, *K. pneumoniae,* and *Candida tropicalis*. MIC values were determined by serial two-fold dilution method, at concentrations ranging from 0.024 mg/mL to 12.5 mg/mL, adjusting final cellular concentration to approximately 5 × 10^5^ cells/mL. The 96-well plates (Orange Scientific, Braine-l’Alleud, Belgium) were incubated at 37 °C for 24–48 h. Sample and cell-free controls were also included. MIC values were determined as the lowest concentrations at which no visible growth was observed. The bactericidal/fungicidal (MBC and MFC) effect of the decoctions was detected by comparison with positive controls (cells grown without extracts). Uninoculated media was used as a negative control to check for sterility. The number of viable cells was assessed by colony-forming unit (CFU) counting after 24 h of incubation at 37 °C. The MBC/MFC values were considered the lowest concentrations at which no CFUs were counted. Experiments were carried out in triplicate and repeated on three independent occasions.

### 4.3. Flow Cytometry

#### 4.3.1. Sample Preparation

An inoculum of *S. aureus* ATCC 25923 was adjusted to an OD_640nm_ equivalent to 1 × 10^6^ cells/mL and kept at 37 °C and 120 rpm. After 4 h of incubation, *Origanum majorana* and *Satureja montana* decoctions were added to a final concentration of 1.56 mg/mL and cells were grown at 37 °C and 120 rpm. Samples of 200 µL were collected after 3, 5, and 24 h of antimicrobials exposure. The cells were centrifuged at 4 °C for 5 min and 9000× *g*, and the pellets were re-suspended in 200 µL of NaCl 0.9%.

#### 4.3.2. Flow Cytometry Analysis

Bacterial viability was assessed using a premixed stain composed of 1.67 mM SYTO 9 and 1.67 mM propidium iodide (PI) (LIVE/DEAD^®^ Baclight™ Bacterial Viability Kit). The premixed stain was added to the prepared samples of *S. aureus* ATCC 25923. The samples were analyzed after 10–15 min of incubation at room temperature in the dark. Flow cytometric analysis was performed on an EC800 Flow Cytometry Analyzer (Sony Biotechnology Inc., Champaign, IL, USA). Both SYTO9 and PI were excited by a diode blue laser (488 nm) and collected through a 530/50 nm bandpass filter in the FL1 channel and 615/30 nm bandpass filter in the FL4 channel, respectively. The sampling rate was 10 µL/min, and the total number of events was set to 40,000 cells. Acquisition was performed using the EC800 software version 1.3.6 (Sony Biotechnology Inc., Champaign, IL, USA) and data analysis, gating, and compensation were carried out using Flow cytometry software FCS Express 6—RUO—version 6.05.0028 (de Novo Software, Glendale, CA, USA). Cells exposed to methanol (100%, 10 min) were used as negative control. This assay was carried out on three different occasions.

#### 4.3.3. Fluorescence Microscopy

To confirm the results obtained by flow cytometry, 10 µL of a stained sample was observed on an Olympus BX51 fluorescence microscope. Photomicroscopy was carried out using the Olympus CellSens imaging software.

### 4.4. Identification and Quantification of Phenolic Compounds

The phenolic profile was determined by LC-DAD-ESI/MSn (Dionex Ultimate 3000 UPLC, Thermo Scientific, San Jose, CA, USA). These compounds were separated and identified, as previously described by Bessada et al. [39]. The obtained extracts were re-dissolved at a concentration of 5 mg/mL prior to analysis. A double online detection was performed using a Diode Array Detector (DAD, 280, 330, and 370 nm as preferred wavelengths) and a mass spectrometer (MS). The MS detection was performed in negative mode, using a Linear Ion Trap LTQ XL mass spectrometer (Thermo Finnigan) equipped with an Electrospray ionization (ESI) source. The identification of the phenolic compounds was performed based on their chromatographic behavior and UV-vis and mass spectra by comparison with standard compounds, when available, and data reported in the literature, giving a tentative identification. Data acquisition was carried out with Xcalibur^®^ data system (Thermo Scientific, San Jose, CA, USA). For quantitative analysis, a calibration curve for each available phenolic standard (apigenin-6-*C*-glucoside, apigenin-7-*O*-glucoside, luteolin-6-*C*-glucoside, *p*-coumaric acid, *p*-hydroxybenzoic acid, quercetin-3-*O*-glucoside, rosmarinic acid, Extrasynthèse, Genay, France) was constructed based on the UV-vis signal. For the identified phenolic compounds for which a commercial standard was not available, the quantification was performed using the calibration curve of the most similar available standard. The results were expressed as milligram/gram of extract.

### 4.5. Statistical Analysis

All samples of decoction were prepared and analyzed in triplicate. The phenolic characterization results were expressed as mean values and standard deviation (SD) and analyzed using a Student’s *t*-test, with α = 0.05, performed with the SPSS v. 23.0 program.

## 5. Conclusions 

Although the specific mechanisms of action of *Satureja montana* and *Origanum majorana* decoction are not yet well understood, new insights into their mode of action were provided in this study. The antimicrobial activity of *Satureja montana* and *Origanum majorana* decoction extracts seems to be attributed to cell membrane perturbation, resulting in the impairment of cell membrane integrity, especially in the first moments after exposure. Overall, this work revealed the bioactive potential of medicinal and aromatic plants showing that the decoctions of *Satureja montana* and *Origanum majorana* are rich in rosmarinic acid and show promise as antimicrobial agents. Further studies are needed to elucidate the in vivo efficacy of these decoctions, to assess their feasibility to be incorporated in formulations for antimicrobial purposes. 

## Figures and Tables

**Figure 1 antibiotics-09-00294-f001:**
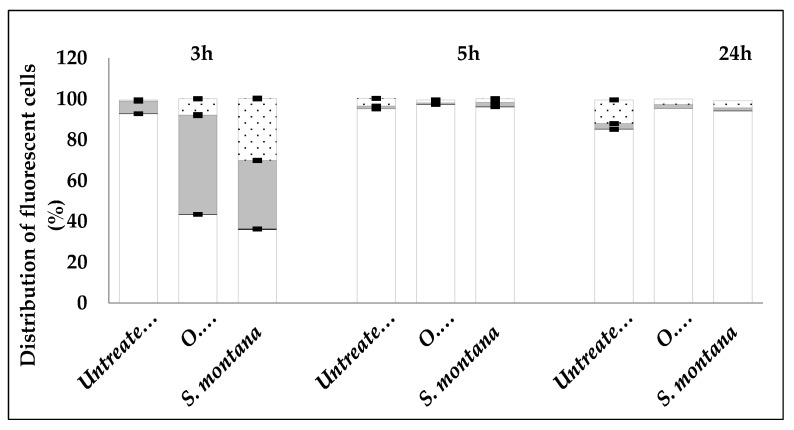
Membrane integrity of *S. aureus* ATCC 25923 after 3, 5, and 24 h exposure to *Satureja montana* and *Origanum majorana* decoctions at 1.56 mg/mL. White bars with dots represent cell fragments or unstained cells; white bars represent cells with intact cell membranes; Gray bars represent cells with slight cell membrane permeabilization; Black bars represent cells with (complete) cell membrane permeabilization. The means ± standard deviations for three independent assays are presented.

**Figure 2 antibiotics-09-00294-f002:**
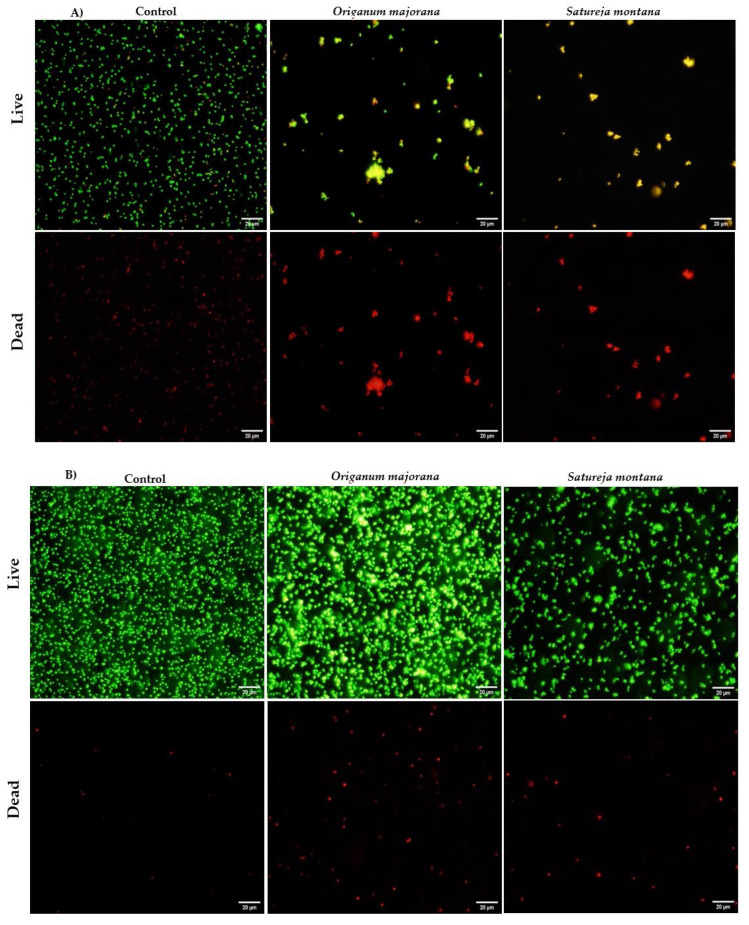
Live/dead cell viability images of *S. aureus* following 3 h (**A**) and 24 h (**B**) exposure to 0 (control—untreated cells) and 1.56 mg/mL of *Origanum majorana* and *Satureja montana* decoctions. Live cells were stained with green and dead cells were stained with red. Magnification 400×. Scale bar = 20 µm.

**Table 1 antibiotics-09-00294-t001:** Antimicrobial activity of *Satureja montana* and *Origanum majorana* decoctions, determined by disc diffusion.

Species	Antimicrobial Activity (Inhibition Zone, mm)	
	*Satureja montana*	*Origanum majorana*
Gram-positive Bacteria		
*Staphylococcus aureus*	+++	+++
*Enterococcus faecalis*	+++	+++
*Streptococcus dysgalactiae*	++	++
Gram-negative bacteria		
*Escherichia coli*	-	-
*Pseudomonas aeruginosa*	++	+++
*Klebsiella pneumoniae*	+++	+++
Yeast		
*Candida albicans*	-	-
*Candida tropicalis*	++	-
*Candida glabrata*	-	-
*Candida parapsilosis*	-	-
Negative control	-	-

(-) absence of halo, (++) moderate halo (8–10 mm), and (+++) strong halo (greater than 11 mm).

**Table 2 antibiotics-09-00294-t002:** Minimum inhibitory concentration (MIC) and minimum bactericidal/fungicidal concentration (MBC/MFC) (mg/mL) of *Satureja montana* and *Origanum majorana* decoctions against *S. aureus*, *K. pneumoniae,* and *C. tropicalis*, determined using the Clinical and Laboratory Standards Institute (CLSI) broth microdilution method.

	*Satureja montana*		*Origanum majorana*	
Antibacterial	MIC	MBC	MIC	MBC
Gram-positive bacterium				
*S. aureus*	1.56	1.56	1.56	1.56
Gram-negative bacterium				
*K. pneumoniae*	1.56	1.56	1.56	1.56
Antifungal	MIC	MFC	MIC	MFC
*C. tropicalis*	6.25	6.25	ND	ND

ND: Not determined.

**Table 3 antibiotics-09-00294-t003:** Retention time (Rt), wavelengths of maximum absorption in the visible region (λmax), mass spectral data, tentative identification, and quantification (mg/g of extract) of the phenolic compounds present in *Satureja montana* and *Origanum majorana* decoctions.

Peak	Rt (min)	λmax (nm)	[M − H]^−^ (*m*/*z*)	MS2 (*m*/*z*)	Tentative Identification	Reference Used for Identification	Quantification (mg/g of Extract)		Student’s *t*-Test *p*-value
							*Satureja montana*	*Origanum majorana*	
1	6.13	310	325	163(100)	*p*-Coumaroyl acid hexoside (A)	[17]	nd	1.21 ± 0.003	-
2	7.15	345	609	489(100),399(10),369(6)	Luteolin-*C*-hexoside-*C*-hexoside (B)	[17,18]	nd	2.20 ± 0.02	-
3	7.26	281	137	93(100)	*p*-Hydroxybenzoic acid (C)	DAD/MS	1.21 ± 0.04	nd	-
4	9.55	336	593	503(30),473(100),383(12),353(21)	Apigenin-6,8-di-*C*-hexoside isomer I (D)	[16,18]	1.7 ± 0.1	7.3 ± 0.2	<0.001
5	9.89	335	593	503(31),473(100),383(11),353(22)	Apigenin-6,8-di-*C*-hexoside isomer II (D)	[16,18]	4.3 ± 0.1	nd	-
6	11.73	346	637	285(100)	Luteolin-*O*-di-glucuronide (E)	[8,19]	nd	3.8 ± 0.2	-
7	12.41	339	637	285(100)	Luteolin-*O*-di-glucuronide (E)	[8,19]	1.5 ± 0.1	2.14 ± 0.02	<0.001
8	14.31	350	477	301(100)	Quercetin-*O*-glucuronide (E)	DAD/MS	1.43 ± 0.02	nd	-
9	14.41	339	623	461(100),285(18)	Luteolin-*O*-hexoside-*O*-glucuronide (E)	DAD/MS	nd	1.58 ± 0.04	-
10	14.57	342	637	461(100),285(15)	Luteolin-*O*-di-glucuronide (E)	[8,19]	1.27 ± 0.01	nd	-
11	15.41	337	799	513(100),285(15)	Luteolin derivative (E)	DAD/MS	nd	6.874 ± 0.002	-
12	15.81	340	623	461(100),285(18)	Luteolin-*O*-hexoside-*O*-glucuronide (E)	DAD/MS	1.36 ± 0.01	nd	-
13	17.25	340	579	285(100)	Luteolin-*O*-pentosyl-hexoside (E)	DAD/MS	nd	1.53 ± 0.01	-
14	17.6	340	461	285(100)	Luteolin-O-glucuronide (E)	[16,17]	3.8 ± 0.1	3.3 ± 0.1	<0.001
15	18.33	340	783	285(100)	Luteolin-*O*-di-glucuronyl-deoxyhexoside (E)	DAD/MS	nd	1.61 ± 0.03	-
16	20.47	325	359	197(28),179(35),161(100)	Rosmarinic acid (F)	DAD/MS; [15,20]	36.3 ± 0.4	52.4 ± 0.2	<0.001
17	23.57	340	461	285(100)	Luteolin-*O*-glucuronide (E)	[16,17]	8.1 ± 0.1	nd	-
18	23.88	288/322	717	537(29),519(100), 493(10),359(10),339(6), 321(6),295(5),197(5), 179(5)	Salvianolic acid B isomer I (F)	[18,19]	0.683 ± 0.001	5.4 ± 0.1	<0.001
19	23.94	326	537	493(50),359(100),313(8),295(2),269(2), 197(29),179(34)	Lithospermic acid A isomer I (F)	[8,21]	16.9 ± 0.2	nd	-
20	27.57	289/323	493	359(100),313(10),295(5), 269(5),197(5),179(5)	Salvianolic acid A isomer I (F)	[21]	20.0 ± 0.7	3.4 ± 0.3	<0.001
21	28.97	289/323	493	359(100),313(10),295(5),269(5),197(5),179)	Salvianolic acid A isomer II (F)	[21]	2.692± 0.003	nd	-
22	30.01	323	537	493(100),359(42), 313(10),295(5),269(5), 197(5),179(5)	Lithospermic acid A isomer II (F)	[8,21]	2.752 ± 0.001	nd	-
23	30.68	327	591	283(100),269(5)	Acacetin-*O*-glucuronide(G)	[16]	4.39 ± 0.03	nd	-
24	32.64	287/321	717	537(29),519(100),493(10),359(10),339(6),321(6), 295(5),197(5),179(5)	Salvianolic acid B isomer II(F)	[18,19]	4.7 ± 0.1	nd	-
Total phenolic acids							85.22 ± 1.03	72.4 ± 0.3	<0.001
Total flavonoids							27.9 ± 0.1	20.4 ± 0.2	<0.001
Total phenolic compounds							113.1 ± 0.9	92.8 ± 0.5	<0.001

nd—not detected. Standard calibration curves: A—*p*-coumaric acid (y = 301950x + 6966.7, R² = 0.9999); B—luteolin-6-*C*-glucoside (y = 4087.1x + 72589, R² = 0.9988); C—*p*-hydroxybenzoic acid (y = 208604x + 173056, R² = 0.9995); D—apigenin-6-*C*-glucoside (y = 107025x + 61531, R² = 0.9989); E—quercetin-3-*O*-glucoside (y = 34843x – 160173, R² = 0.9998); F—rosmarinic acid (y = 191291x – 652903, R² = 0.999); G—apigenin-7-*O*-glucoside (y = 10683x – 45794, R² = 0.9906).

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
