# Peer review of "Satureja montana* L. and *Origanum majorana* L. Decoctions: Antimicrobial Activity, Mode of Action and Phenolic Characterization"

_antibiotics, 2020, doi:10.3390/antibiotics9060294_

Round 1

Reviewer 1 Report

I thank the authors for their response however a number of issues remain.

  1. There continues to be an issue with the use of reference material in this work. For example, the statement at line 37-38 is not supported by references 7 & 8, further, I’m not sure that there is evidence to support the claim that self-medication with plants is safer practice.  Similarly, references 6 & 9 do not support the claim made in the respective sentences. This problem also occurs in the discussion where references 11, 12, 21-33 are used to support a statement about essential oils however several of these studies not investigate essential oils. I recommend that all cited work is double checked to ensure that the reference provides evidence for what is claim.
  2. Line 58 describes strong activity as 8-33 mm however in Table 1 and the methods strong activity is > 11 mm. Based on this activity of S. montana against P. aeruginosa and both extracts against are moderate not strong.
  3. While the previous statement regarding data not shown has been removed the authors have now included this statement for the data in Table 1 (line 58-59).
  4. It is also unclear why the authors, having measured zones, have now chosen to display the data as +, ++ etc. I appreciate that this was an approach I raised in my review however this should have been a decision made in the experimental design phase. Given that the authors have exact values statistical evaluation could easily have been used to analyse this data. I note however that there is no indication in the Methods as to how many replicates were done which raises the question of whether this is from a single data point.
  5. No positive or negative control information is provided in Table 2, or positive control data for Table 1.
  6. It is unclear why only 3 organisms were included for MBC and MIC assays, while the authors’ state that only those with the highest activity were included based on the information in Table 1 E. faecalis and P. aeruginosa should have been included. This problem seems to arise from the authors changing the analysis approach after key experimental design decisions were made.
  7. Figure 2 needs the inclusion of scale bars. In this figure it is unclear what the labels live and dead mean on the left-hand side of the image.
  8. There are some grammatical errors in the work, for example, incorrect capitalisation in organism names.
  9. Table 3 – units are needed for the quantification data within the column heading.

Author Response

Reviewer 1:

Each point raised by the reviewer 1 was taken into account and corrected accordingly (corrections highlighted in grey in the manuscript).

  1. There continues to be an issue with the use of reference material in this work. For example, the statement at line 37-38 is not supported by references 7 & 8, further, I’m not sure that there is evidence to support the claim that self-medication with plants is safer practice. Similarly, references 6 & 9 do not support the claim made in the respective sentences. This problem also occurs in the discussion where references 11, 12, 21-33 are used to support a statement about essential oils however several of these studies not investigate essential oils. I recommend that all cited work is double checked to ensure that the reference provides evidence for what is claim.

As recommended by the reviewer, all references cited were checked. However, some of those mentioned by the reviewer were maintained as justified below.

Introduction: “Medicinal plants and their phenolic compounds are thus strong candidates for the development of new drugs [5], having the advantages of worldwide availability, low price and safety [6].”

“Hence, finding safe and effective therapeutic approaches to prevent and treat such diseases, particularly with the use of medicinal plants, which are safer, cheaper, and more available, is an important goal that we are seeking to achieve with the help of novel and fast cheminformatics methods [10] in the present study.” From Pour et al., 2017 [6]. The reference of Majumder et al., 2020 was added to the manuscript.

 “From this review, it is clear that plant-derived phytochemicals represent a possible source of effective, cheap and safe antimicrobial agents.” (Majumder et al., 2020)

“In this context, extracts from plant origin are a promising antimicrobial alternative worthy of further exploration [9].” In this study Martins et al. studied the antimicrobial activity of extracts of several plant (commonly used in folk medicine) against several Candida spp. concluding that “Overall, hydromethanolic plant extracts could constitute promissory alternatives to the traditional antifungal agents.” However, the authors stressed " the need of further studies to elucidate the mechanisms of action of the tested plant extracts, as well as to test their in vivo efficacy”. Other references supporting this sentence were included.

Discussion: “The antimicrobial potential of plant species has been explored mostly via studies of their essential oils (EOs) [11,12,21-33].” This sentence is supported by the cited references, since it is said that most studies focus on the study of essential oils and therefore in our opinion support the claim made.

  1. Line 58 describes strong activity as 8-33 mm however in Table 1 and the methods strong activity is > 11 mm. Based on this activity of S. montana against P. aeruginosa and both extracts against are moderate not strong.
  2. While the previous statement regarding data not shown has been removed the authors have now included this statement for the data in Table 1 (line 58-59).
  3. It is also unclear why the authors, having measured zones, have now chosen to display the data as +, ++ etc. I appreciate that this was an approach I raised in my review however this should have been a decision made in the experimental design phase. Given that the authors have exact values statistical evaluation could easily have been used to analyse this data. I note however that there is no indication in the Methods as to how many replicates were done which raises the question of whether this is from a single data point.

We agree with the reviewer and in fact, the presentation of the results obtained by disc diffusion assay were confusing due to the changes performed during the review process. It is also true that the semi-quantitative scale allows to distinguish between moderate and strong activity and this was corrected in the manuscript. What we wanted to say was that the extracts presenting a halo higher than 8 mm showed good antimicrobial activity. The scale used to score the inhibition of the plant extracts tested is based on the diameter of the halo and used in several studies and allows to score inhibition as absent, weak, moderate and strong. Taking into account the use of this scale and based in the actual version of the results presented in table 1, this part of the result section was rewritten aiming to clarify the results obtained by disc diffusion assay.

  1. No positive or negative control information is provided in Table 2, or positive control data for Table 1.

Positive and negative controls are routinely performed in the laboratory when performing these types of tests. However, as they had no effect, their results were not placed in the respective tables. MIC/MBC: Positive control (untreated cells), negative control (medium without cells). Disk diffusion assay: positive and negative controls (untreated cells and cells exposed to the solvent used to dissolve the extracts -water).

  1. It is unclear why only 3 organisms were included for MBC and MIC assays, while the authors’ state that only those with the highest activity were included based on the information in Table 1 E. faecalis and P. aeruginosa should have been included. This problem seems to arise from the authors changing the analysis approach after key experimental design decisions were made.

For MIC and MBC assays, a Gram positive and a Gram-negative bacterium, and a fungus were chosen and this information was mentioned in the text. Therefore, one representative species of each type of microorganism was chosen, preferably the one with the highest activity and the most often associated with microbial infections. (Gram-positive: S. aureus: high activity and often associated with microbial infection; Gram negative: K. pneumoniae: high activity (both extracts) and C. tropicalis (the only Candida strain with activity).

  1. Figure 2 needs the inclusion of scale bars. In this figure it is unclear what the labels live and dead mean on the left-hand side of the image.

The scale bars were placed in the images of figure 2 and legend. The meanings of live and dead labels were explained/claridied in the text and in the figure caption.

  1. There are some grammatical errors in the work, for example, incorrect capitalisation in organism names.

The grammatical errors mentioned were corrected.

  1. Table 3 – units are needed for the quantification data within the column heading.

The units of the phenolic compounds quantification were placed in the respective column of table 3.

Reviewer 2 Report

"These assays also demonstrate that both plant extracts acted by perturbing the cell membrane structures and architecture, especially in the first moments upon exposure.!

This statement must be supported by other experimental evidence or be put as an hypothetical mechanism of action.

Figure 1 shall be redone to show standard deviation bars and post hoc test.

Author Response

Reviewer 2:

Each point raised by the reviewer 2 was taken into account and corrected accordingly (corrections highlighted in blue in the manuscript).

"These assays also demonstrate that both plant extracts acted by perturbing the cell membrane structures and architecture, especially in the first moments upon exposure.!

This statement must be supported by other experimental evidence or be put as a hypothetical mechanism of action.

This sentence was changed according to the reviewer's suggestions.

Figure 1 shall be redone to show standard deviation bars and post hoc test.

Figure 1 was formatted in order to make the error bars more noticeable.

Reviewer 3 Report

The manuscript describes the preparation of decoctions of the leaves of two plants (Satureja montana L. and Origanum majorana L.) used in folk medicine, as well as in cooking. The antimicrobial activity of decoctions in vitro against a number of bacteria and fungi of the genus Candida spp. was determined. The main components of the compositions were analyzed by HPLC. MICs were got for decoctions that were active against Gram-positive bacteria and one fungus. Using fluorescence microscopy, it was shown that the mechanism of action of bactericidal substances is associated with a violation of the integrity of the cell membrane.

The work was done in good faith, relevant literature on this topic is provided. Decoctions from this work are compared with published data on plant extracts. The authors consider their results as material for further research. It would be important to understand which substances are responsible for the antimicrobial effect and whether there is a synergistic effect.

The work may be published, but requires corrections. Basically, corrections relate to a clearer presentation of the results and correction of typos (see table below).

Line

Now in text

Please fix it

17

charactrization

characterization 

92 Fig.1

O. Majorana

O. majorana

92 Fig.1

Black bars represent cells with (complete) cell membrane permeabilization. The means ± standard deviations for three independent assays are illustrated.

Black columns are indistinguishable from column borders. Standard deviation is not clear as shown.

44-46

O. majorana is a perennial plant originally native of the Mediterranean region, but widely cultivated in many countries S. montana is a plant native from the Mediterranean region and cultivated all over Europe, Russia and Turkey [10,11].

Please edit

58

33 mm of diameter(data

33 mm of diameter (data

60

Candida spp.

Candida spp.

Table 1

Staphylococcus Aureus

Enterococcus Faecalis

Staphylococcus aureus

Enterococcus faecalis

Figure 2: Fluorescence microscopy (LIVE/DEAD® Baclight™ Bacterial Viability Kit; Life Technologies) of

In my version of the text (PDF), the designations of the figures are shifted. The meaning is clear only from the text of the manuscript.

Please correct the captions to all parts of the drawing.

Table 2

Gram-positive bacteria

Gram-negative bacteria

Gram-positive bacterium

Gram-negative bacterium

208

The leaves of Satureja montana L. (winter savory) and Origanum majorana L. (marjoram) were

Dry leaves?

300

decoctions of S. montana and O. majorana are rich in rosmarinic acid

S. montana and O. majorana

S. montana

S. dysgalactiae

S. aureus

S. parvifolia

To representatives of different genera of organisms should be given different reductions.

Author Response

Reviewer 3:

Each point raised by the reviewer 3 was taken into account and corrected accordingly (corrections highlighted in yellow in the manuscript).

Line 17: charactrization was replaced by characterization;

Line 92 fig 1: O. Majorana was replaced by Origanum majorana.

Line 44-46: O. majorana is a perennial plant originally native of the Mediterranean region, but widely cultivated in many countries S. montana is a plant native from the Mediterranean region and cultivated all over Europe, Russia and Turkey [10,11].

This sentence was edited.

Line 58: 33 mm of diameter(data was replaced by 33 mm of diameter (data

Line 60: Candida spp. was replaced by Candida spp.

Table 1: Staphylococcus Aureus and Enterococcus Faecalis were replaced by Staphylococcus aureus and Enterococcus faecalis, respectively.

Table 2: Gram-positive bacteria and Gram-negative bacteria was replaced by Gram-positive bacterium and Gram-negative bacterium, respectively.

 Line 208: The leaves of Satureja montana L. (winter savory) and Origanum majorana L. (marjoram) wereDry leaves? This information was introduced in the text.

Line 300: decoctions of S. montana and O. majorana are rich in rosmarinic acid was replaced by decoctions of Satureja montana and Origanum majorana are rich in rosmarinic acid.

  1. montana, S. dysgalactiae, S. aureus, S. parvifolia. To representatives of different genera of organisms should be given different reductions.

In order to distinguish between plants and bacteria, we chose to keep the abbreviation of bacteria (ex: S. aureus) and write in full the genus name of plants (ex: Satureja montana, Origanum majorana, Satureja parvifolia, etc).

Figure 2: In my version of the text (PDF), the designations of the figures are shifted. The meaning is clear only from the text of the manuscript. Please correct the captions to all parts of the drawing.

The final version of the manuscript (after converting to pdf) will be carefully checked for this issue.

Line 92 Figure 1: Black bars represent cells with (complete) cell membrane permeabilization. The means ± standard deviations for three independent assays are illustrated. Black columns are indistinguishable from column borders. Standard deviation is not clear as shown.

Figure 1 was formatted in order to make the black bars and error bars more noticeable. However, it is necessary to take into account their low values, which makes it difficult to visualize.

Reviewer 4 Report

Line 56: is it possible to add (maybe even as supplemental data) the actual results, even if representatives? So the reader can estimate what is the difference between strong effect to weak effect as the results presented are not quantitative.

Alternatively, the author may use values of diameter. The author wrote for this data “not shown” I think it is necessary.

From the abstract, I was expecting that the paper will present microbial activity at the level of cell membranes. It would help to have a short introduction at the beginning of section 2.2, on how analysis of flow cytometry and fluorescent microscopy is used for the determination of membrane integrity.

In Table 2, statistical analysis is required (standard deviation for the reported values).

Line 90: Figure legend, I believe the author meant presented (instead of “illustrated”).

Line 114: Figure legend 2 is not explanatory. Figure 2 is difficult to follow (both from the text and the legend). I suggest to rearrange the figure and re-write the text explaining the figure.

The conclusions section is too broad and can be improved, by elaborating on the mechanism of action, and maybe on the identified of phenolic compounds and their plausible role (especially those who presented high quantities).

Author Response

Reviewer 4:

Each point raised by the reviewer 4 was taken into account and corrected accordingly (corrections highlighted in green in the manuscript).

Line 56: is it possible to add (maybe even as supplemental data) the actual results, even if representatives? So the reader can estimate what is the difference between strong effect to weak effect as the results presented are not quantitative.

Alternatively, the author may use values of diameter. The author wrote for this data “not shown” I think it is necessary.

The results of the disk diffusion assay obtained were converted in a semi-quantitative scale and inhibition scored as absent, weak, moderate and strong, depending on the size of the halos (-) absence of halo, (+) weak halo (3-7mm), (++) moderate halo (8-10mm) and (+++) strong halo (greater than 11mm). In order to better understand the results obtained, they were rewritten.

From the abstract, I was expecting that the paper will present microbial activity at the level of cell membranes. It would help to have a short introduction at the beginning of section 2.2, on how analysis of flow cytometry and fluorescent microscopy is used for the determination of membrane integrity.

 A short introduction was introduced in the section 2.2. as suggested by the reviewer.

In Table 2, statistical analysis is required (standard deviation for the reported values).

 The MIC values were determined by Two-fold Broth Microdilution Method. The values presented are the results of experiments carried out in triplicate and repeated in three independent assays. However, in some cases where dubious results were obtained (since it is a visual test), another assay was carried out to obtain coherent and conclusive results. Thus, the same value of MIC and MBC was obtained in the three independent assays presented in table 2.

Line 90: Figure legend, I believe the author meant presented (instead of “illustrated”).

Illustrated was replaced by presented.

Line 114: Figure legend 2 is not explanatory. Figure 2 is difficult to follow (both from the text and the legend). I suggest to rearrange the figure and re-write the text explaining the figure.

The figure legend and the text were rewritten according to the reviewer's comment.

The conclusions section is too broad and can be improved, by elaborating on the mechanism of action, and maybe on the identified of phenolic compounds and their plausible role (especially those who presented high quantities).

The conclusions were rewritten according to the reviewer's suggestions. However, currently, we already know, (assays testing rosmarinic acid were performed in order to evaluate their possible role as active component), that although being the most abundant phenolic compound and being present in a higher concentration in both extracts (32% of the total phenolic compounds in S. montana and 56% in O. majorana), rosmarinic acid alone was not the active compound responsible for the antimicrobial activity exhibited by the plant extracts studied. As observed by other authors the abundance of a phenolic compound is not always a sign of bioactivity. As suggested by Martins et al. (2016) vestigial molecules may contribute to the bioactivity of plant extracts as well as to be responsible for the activation of other active molecules (synergy/additivity) (Martins et al. 2016). Thus, we opted by not concluding anything about the plausible role of rosmarinic acid.

Martins, N.; Ferreira, I.C.F.R.; Henriques, M.; Silva, S. In vitro anti-Candida activity of Glycyrrhiza glabra L. Ind. Crops Prod. 2016, 83, 81–85. https://doi.org/10.1016/j.indcrop.2015.12.029.

Round 2

Reviewer 1 Report

I thank the authors for their responses, this has addressed some of my comments however the following issues remain:

  1. While the authors’ claim to have checked that citations are appropriate there are still issues. For example references 6 and 7 are used to support a statement about availability, cost and safety of plant extracts however neither reference addresses these areas. In addition the data provided for reference 7 is incorrect – this work was published in 2019 not 2020. I have not checked other references for accuracy however that this issue remains, especially after raising this previously, raises concern that other citations may not be accurate.
  2. I am also concerned that a statement regarding further exploration of plant extracts only cites studies by the authors and represents antimicrobial studies on a small number of plants. There is a large body of literature on this topic, these works look at this matter in a much broader and comprehensive way. The references provided (ref 8-10) do not speak to the statement at line 38-38.
  3. Figure 2 – while scale bars have been added they are almost invisible in the images as they are too small and lack contrast against the image background.
  4. Figure 2 – what do T3h and T24h mean in the image? I assume this is the control however this is not clear.
  5. Table 3 – there is inconsistent presentation of data in this table with numbers being presented to 1, 2 and 3 decimal places.
  6. Use of positive and negative is addressed in the authors’ comments however this data (i.e. results) is not presented and there does not seem to have been any positive control used for the disc diffusion assay.

Author Response

Reviewer 1:

  1. While the authors’ claim to have checked that citations are appropriate there are still issues. For example references 6 and 7 are used to support a statement about availability, cost and safety of plant extracts however neither reference addresses these areas. In addition the data provided for reference 7 is incorrect – this work was published in 2019 not 2020. I have not checked other references for accuracy however that this issue remains, especially after raising this previously, raises concern that other citations may not be accurate.

Answer: “Medicinal plants and their phenolic compounds are thus strong candidates for the development of new drugs [5], having the advantages of worldwide availability, low price and safety [6,7].”

In this sentence, taken from the article, reference 6 and 7 aims to support the entire sentence. In reference 6, the authors address the need of find “safe and effective therapeutic approaches particularly with the use of medicinal plants,” that according to them “are safer, cheaper, and more available.” The reference 7 is a review that addresses the use of plant extracts and phytochemicals as potential source of antimicrobial agents. In this review, the authors explored “the prospects and potentials of phytochemicals as an effective alternative source of effective, cheap and safe antimicrobial agents.” and concluded that “plant extracts can be an attractive potential source for future antibacterial drugs.” and that “… plant-derived phytochemicals represent a possible source of effective, cheap and safe antimicrobial agents.” Thus, in our opinion both references support the claim made in the sentence in question.

Regarding the reference 7 (DOI: 10.20959/wjpps20201-15237), the information that we found online referred that it was accepted on 1 January 2020 and published in 2020 (https://www.researchgate.net/publication/338549080_EXPLORING_PHYTOCHEMICALS_AS_ALTERNATIVES_TO_ANTIMICROBIALS-PROSPECTS_AND_POTENTIALS#fullTextFileContent)

  1. I am also concerned that a statement regarding further exploration of plant extracts only cites studies by the authors and represents antimicrobial studies on a small number of plants. There is a large body of literature on this topic, these works look at this matter in a much broader and comprehensive way. The references provided (ref 8-10) do not speak to the statement at line 38-38.

Answer: Although the reviewer is correct and there are much more literature in the topic, the authors thought that the references selected for the document are representative of the studies that support the phrase in question.

Statement of line 38: “Now, the rising levels of resistance to traditional antimicrobial therapies is widely recognized as a global issue and requires the urgent development of new approaches to fight microbial infections.” This sentence is authored by the authors and, in our opinion, needs no reference, since antimicrobial resistance and the urgent need for alternatives is a well-known issue.

  1. Figure 2 – while scale bars have been added they are almost invisible in the images as they are too small and lack contrast against the image background.

Answer: The scale bar was formatted in order to make it more noticeable.

  1. Figure 2 – what do T3h and T24h mean in the image? I assume this is the control however this is not clear.

Answer: T3h and T24 h are the exposure time to plant extracts. After 3 and 24 h of incubation a sample of the control (untreated cells), and cells exposed to S. montana and O. majorana were collected and studied by flow cytometry using live/dead assay. For each time exposure, dead cells are stained red and live cells are stained green. This issue was clarified in the figure and respective legend.

  1. Table 3 – there is inconsistent presentation of data in this table with numbers being presented to 1, 2 and 3 decimal places.

Answer: The number of significant digits of the data and the terms used in the tables were double-checked as suggested. However, it should be noted that the number of significant digits is conditioned by the standard deviation (SD). SD is a statistical calculation that is a measure of how much scatter (or uncertainty) there is in the data. If we round the SD to one significant digit that will tell us in which decimal place the uncertain digit of our final result lies. Therefore, for experimental results, determining significant digits this way is a more reliable way of reporting precision because it takes experimental error into account and is not based solely on the initial measurements themselves. This way of expressing the results can be found in statistical books and scientific articles.

  1. Use of positive and negative is addressed in the authors’ comments however this data (i.e. results) is not presented and there does not seem to have been any positive control used for the disc diffusion assay.

Answer: In fact, the positive control was not referred for the halo test, since it is the bacterial/fungal growth on the plate. If bacteria/fungi grow on the rest of the plate it already represents the positive control, and therefore we thought that there was no need to refer this in the manuscript, but if the reviewer think it should appear there we can add it.

Round 3

Reviewer 1 Report

I regret that the authors’ response to my previous comments is not satisfactory.

  1. In appropriate use of citations – while the authors claim that they have checked the sources to ensure that the claims made in the paper are supported by the reference this is not the case. For example reference 7 is used to support a claim of safe, low cost and available treatments however reference 7 does not provide any evidence for this. Similar reference 6 does not provide any comment on these aspects as it is a report of an in vitro study of anticoagulant activity. These are just two examples and there are several others, for example use of reference 8 and 9 to support a statement about historical use of medicinal plants. These works are also the work of the authors and this seems to be a clear example of inappropriate self-citation.
  2. I accept the authors comment regarding the publication date.
  3. The authors have not addressed the issue of high level of self-citation. Of the 39 works cited in this paper 8 are by the authors (20.5%) and in several places there many works in the literature that more directly address, or provide evidence for, the statement that is being made.
  4. The scale bars have been improved.
  5. Table 3 - Aside from the issue of consistency of data presentation the data presented in Table 3 has all been measured within the same assay and therefore all raw data has the same level of imprecision. To then present some data with more precision than others is misleading and known as false precision. There is a large body of literature on this topic and the authors are recommend to consult this information and how to present means and standard deviation (see for example https://www.ncbi.nlm.nih.gov/pmc/articles/PMC4483789/) .
  6. The response by the authors suggests that they do not understand the difference between positive and negative controls. In a disc diffusion assay the negative control occurs when a disc with solvent only (i.e. no extract) is used, for a positive control a treatment of known response, typically a standard antibiotic, is used. Both control types are essential and ensure that the experimental system is operating as expected and hence results from extracts with unknown activity can be appropriately interpreted.